# Intra-Arterial Delivery of Radiopharmaceuticals in Oncology: Current Trends and the Future of Alpha-Particle Therapeutics

**DOI:** 10.3390/pharmaceutics15041138

**Published:** 2023-04-04

**Authors:** Nathan Kauffman, James Morrison, Kevin O’Brien, Jinda Fan, Kurt R. Zinn

**Affiliations:** 1Comparative Medicine and Integrative Biology, Institute for Quantitative Health Science and Engineering, Michigan State University, East Lansing, MI 48824, USA; kauffm57@msu.edu; 2Advanced Radiology Services, 3264 N Evergreen Dr, Grand Rapids, MI 49525, USA; jmorrison@advancedrad.com; 3Department of Radiology, Henry Ford Health System, Detroit, MI 48202, USA; kevino@rad.hfh.edu; 4Departments of Radiology and Chemistry, Institute for Quantitative Health Science and Engineering, Michigan State University, East Lansing, MI 48824, USA; fanjinda@msu.edu; 5Departments of Radiology, Biomedical Engineering, Small Animal Clinical Sciences, Institute for Quantitative Health Science and Engineering, Michigan State University, East Lansing, MI 48824, USA

**Keywords:** cancer, imaging, therapy, interventional radiology

## Abstract

A paradigm shift is underway in cancer diagnosis and therapy using radioactivity-based agents called radiopharmaceuticals. In the new strategy, diagnostic imaging measures the tumor uptake of radioactive agent “X” in a patient’s specific cancer, and if uptake metrics are realized, the patient can be selected for therapy with radioactive agent “Y”. The X and Y represent different radioisotopes that are optimized for each application. X–Y pairs are known as radiotheranostics, with the currently approved route of therapy being intravenous administration. The field is now evaluating the potential of intra-arterial dosing of radiotheranostics. In this manner, a higher initial concentration can be achieved at the cancer site, which could potentially enhance tumor-to-background targeting and lead to improved imaging and therapy. Numerous clinical trials are underway to evaluate these new therapeutic approaches that can be performed via interventional radiology. Of further interest is changing the therapeutic radioisotope that provides radiation therapy by β- emission to radioisotopes that also decay by α-particle emissions. Alpha (α)-particle emissions provide high energy transfer to the tumors and have distinct advantages. This review discusses the current landscape of intra-arterially delivered radiopharmaceuticals and the future of α-particle therapy with short-lived radioisotopes.

## 1. Introduction

Radiation therapy is a pillar of oncological care. External-beam radiation therapy is a well-established modality of tumor therapy that has gone through many technological advancements to improve accuracy, safety, and efficiency. Even with these advancements, the external beam has limitations, such as treating tumors near sensitive or mobile organs, deep-seated tumors, or wide-spread metastases [1,2,3]. Efforts to deliver radiation treatment in these situations have led to alternative radiotherapy technologies. Brachytherapy and Selective Internal Radiation Therapy (SIRT) place radioactive sources or drugs near or within tumors [4,5,6]. An emerging alternate approach is the cancer-targeted intravenous delivery of small peptides or antibodies that are radiolabeled with beta (β)-emitters to address situations where an external beam is contraindicated. Variations on this theme include a “pretargeting” approach where an unlabeled antibody–avidin conjugate is administered and allowed to “pretarget” tumors over multiple days, followed by injection of radiolabeled biotin that quickly binds the pretargeted conjugate in the tumor, with any unbound radiolabeled biotin rapidly cleared to minimize the radiation dose to normal tissues [7,8,9,10]. This original strategy has been extended to include click chemistry instead of the avidin–biotin approach and can also be applied to cross the blood–brain barrier [11,12]. Certain thyroid cancers express a specific transporter for iodide that can be exploited for imaging and therapy. One of the oldest and most accepted treatments for thyroid cancer uses I-123-iodide for SPECT imaging and dosimetry, followed by I-131-iodide for radiation therapy [13,14]. This concept of imaging first, then selecting the appropriate therapy, has evolved into “radiotheranostics” and is rapidly advancing by using novel targeting strategies and new therapeutic radioisotopes. The purpose of this review is to discuss the current landscape of radiotheranostics, which has recently been FDA-approved for intravenous injection for imaging and therapy but is now under intensive evaluation for intra-arterial (IA) dosing. In addition, the potential of new radioisotopes that decay by α-particle emissions will be explored.

## 2. Radiotheranostics

Theranostics is a modern term that refers to the use of a diagnostic agent first to identify patients with targetable diseases, followed by treating those same patients with a therapeutic agent that is analogous to the diagnostic agent [15]. In nuclear medicine applications for cancer, the agent can be referred to as a radiotheranostic. It typically has a chelator molecule that binds one radioisotope for imaging and a second radioisotope for the therapeutic treatment of the cancer [16]. This concept is shown in Figure 1. The same chelator may be used to tightly bind both the imaging radioisotope and therapeutic radioisotope separately, or different chelators may be used for each. This strategy allows for an initial imaging session with Positron Emission Tomography (PET), Single-Photon Emission Computed Tomography (SPECT), or gamma camera imaging to confirm the high uptake or proper targeting of the agent in an individual patient’s tumor before proceeding to a therapeutic radioisotope in a later treatment session. The main pillar of radiotheranostics, which involves the use of a radioisotope with a lower radiation dose for screening before committing to the desired therapeutic radioisotope, is not a new concept and has been used widely in nuclear medicine. This is highlighted in Table 1 and Table 2, which summarize FDA-approved radiopharmaceuticals.

The two drug combinations that match the modern definition of radiotheranostics are Gallium-68-DOTATATE/Lutetium-177-DOTATATE (Netspot/Lutathera) and Ga-68-PSMA-11/Ga-68-PSMA-617 (Ga 68 PSMA-11/Pluvicto). Netspot/Lutathera utilizes DOTATATE to target somatostatin receptor-positive neuroendocrine tumors. PET imaging with Ga-68-DOTATATE confirms targeting and high levels of the agent in the tumors to subsequently justify multiple cycles of Lu-177-DOTATATE for β-radiation therapy. Lutathera was approved by the FDA in 2018 after the impressive success of the Phase 3 NETTER-1 trial, where progression-free survival at 20 months post-treatment was 65.2% in the Lu-177-DOTATATE arm and 10.8% in the standard-therapy arm [17].

Following somatostatin-receptor imaging/therapy very closely was the targeting of the prostate-specific membrane antigen (PSMA) for the diagnosis of castration-resistant prostate cancers. Ga-68-PSMA-11 (Ga 68 PSMA-11) was FDA-approved in 2020, followed by F-18-piflufolastat (Pylarify) in 2021. Subsequently, Lu-177-PSMA-617 (Pluvicto) was approved in March 2022 for therapeutic applications [18,19]. PSMA-617 targets PSMA and becomes internalized into cancer cells [20]. Major side effects can result from this therapy due to the off-targeting of the molecule; these include xerostomia, renal damage, and bone marrow ablation. An improvement in total body distribution and the targeting of tumors can not only decrease these side effects but also lead to a better tumor response.

Nearly all current FDA-approved radiopharmaceutical drugs for imaging and therapeutic applications are required to be injected intravenously per their label indications, except for I-123 and I-131 (oral). These cancer-seeking drugs have a biological half-life long enough to bind and accumulate in tumor tissue with sufficient clearance to provide appropriate tumor levels over the background for imaging and to reduce toxic effects on normal tissues [21]. The few radiotheranostic agents available and the narrow range of applicable tumor indications for each drug highlight the challenges in developing them and obtaining FDA approval and insurance reimbursement [22,23,24]. Radiotheranostics have the potential for “off-label” routes of administration under defined conditions, such as during approved clinical trials. Alternate forms of delivery could increase the specificity of targeting agents and increase the tumor-to-healthy-tissue dosing profile [25,26]. Image-guided IA delivery is an established method for a variety of cancer drug therapies, including radioisotope therapy such as yttrium-90, and would thus be applicable for radiotheranostics.

## 3. Interventional Oncology

Interventional oncology (IO) encompasses a variety of minimally invasive procedures utilizing fluoroscopy, ultrasound, or Computed Tomography (CT) imaging guidance to deliver local therapy to tumors [27]. IO procedures are performed by a subspecialized physician who is board certified in Interventional Radiology (IR). Figure 2 shows a typical IR angiosuite and an overview of a Y90 radioembolization procedure. The two main treatment options in IO are ablation and embolization. Ablation consists of tumor destruction achieved through direct medication injection, heating, freezing, or electroporating the tumor tissue. Embolization is a procedure that targets tumors via their arterial blood supply. After navigating a catheter into the appropriate artery, a variety of treatment agents can be directly injected into a tumor. Embolization particles can be mixed with chemotherapy, contain radioactive sources, or simply block the blood supply to the tumor to cause tissue ischemia and destruction. These IO therapies can be used individually, in combination, or with other oncologic treatments to achieve the desired goal.

Patients who are candidates for IO procedures typically have tumors that are limited in size and/or spread of disease. In most oncologic centers, patients are evaluated by a multidisciplinary group of physicians, also known as a “Tumor Board”, to determine the optimal treatment pathway. In many scenarios, IO procedures can be curative. Interventional oncologic procedures can also be utilized to decrease a patient’s tumor burden and make them a candidate for other curative treatments such as surgical resection or transplantation. In the absence of a curative treatment option, IO procedures can also be employed for palliative purposes to prolong life and treat malignancy-related pain and other symptoms. IO therapies are minimally invasive and well tolerated, with most performed under sedation on an out-patient basis. This allows for a larger population of patients to be treated compared to more invasive alternatives.

## 4. Radioembolization

One of the most common IA cancer therapies is the delivery of radioactive, embolic particles directly into tumors, also known as SIRT or Transarterial Radioembolization (TARE). Currently, two types of particles are FDA-approved for use in this procedure: SIR-sphere and Therasphere. Both utilize small embolic beads labeled with yttrium-90 to lodge within tumor-specific arterial vasculature and deliver a therapeutic dose to a tumor with β-radiation. As both these particles use yttrium-90 as the therapeutic agent, the treatment is also sometimes referred to as Y90. The only FDA-approved indication for Y90 is unresectable liver cancer, which can be primary hepatocellular carcinoma or liver metastases from a different primary tumor location [28].

The clinical trial that led to SIR-sphere approval was a phase 3 randomized trial comparing the responses of patients with colorectal cancer metastases to the liver treated with either chemotherapy alone or a combination of SIRT and chemotherapy. Improvements were seen in partial and complete responses, tumor volume, and serum levels of the carcinoembryonic agent [29]. Multiple trials have since confirmed the long-term efficacy of radioembolization, which has already been well summarized [30]. There are many current trials exploring the advancements in radioembolization, including Y90 combination therapies and improved Y90 dosimetry [31,32,33,34]. Two other areas of interest are the radioembolization of extrahepatic tissues and the development of novel radioembolics.

Mouli et al. performed an initial study in dogs to explore using Y90 in the prostate [35]. The study was performed in 14 male castrated beagles with induced prostatic hyperplasia. The treated dogs showed a significant decrease in the treated hemigland compared to the untreated contralateral hemigland control. They also showed no clinical signs of toxicity. Further, histology showed radiation-induced cell death in the treated prostate gland tissue, while radiography and gross observation of the extraprostatic organs showed no changes. This result indicates that the prostate, although a very different tissue compared to the liver, is amenable to Y90. The success of future therapeutic studies could open the door for additional targets, such as breast cancer, which has been shown to be targetable with IA-delivered chemotherapy [36].

Other additional radioembolics being investigated are Iodine-131 lipiodol, Rhenium-188 lipiodol, Rhenium-188 microspheres, and Ho-166 microspheres [28]. In Europe, Ho-166-labeled microspheres are emerging as a third option next to SIR-spheres and Theraspheres [37]. Ho-166 is an attractive option mainly for its partial gamma (γ) emission (6.7%). While pure β-emitters are typically seen as the most ideal therapeutic agent, having some γ emission, such as in the case of Ho-166, allows for better imaging and thus tracking of the microspheres when compared to Y90. Better tracking is available for both the scout imaging session before therapy and the therapy delivery itself. Due to Y90′s poor imaging qualities, technetium-99m-labeled macroaggregated albumin (MAA) is used for scouting and calculating liver shunts to the venous system. MAA is similar in size to Y90 spheres but is not a perfect predictor for Y90 distribution. Since Ho-166 has inherent imaging properties, it can be used at a low dose to predict its own deposition profile and dosimetry upon therapeutic delivery. Post-treatment, Ho-166 deposition can be more accurately imaged with SPECT/CT compared to Y-90 since Y-90 does not have γ emissions but relies on Bremsstrahlung radiation instead, which is not ideal.

Overall, radioembolics are a promising prospect because they are receptor-agnostic, which means they do not rely on the high expression of a cancer-specific biological marker relative to healthy tissues to achieve therapeutic effect. Since cancers are highly heterogeneous across both tissue types and individual patients, having a diverse range of tools is highly beneficial. A recent review of reports investigating new β-emitting microparticles highlights the interest in developing new radioembolics at the preclinical level [38]. Alternatively, combining the specificity of selective IA delivery with cancer receptor binding may prove to be the best option when available.

## 5. Intra-Arterial Delivery of Radiopeptides

The IA delivery of receptor-targeted radiopharmaceuticals has also gained interest at the clinical level [39]. Radiotheranostics delivered via IA administration have the potential to both increase the amount of targeted cancer binding and decrease the uptake in non-target tissues, thereby reducing toxicity. Although Lu-177-DOTATATE (Lutathera) and Lu-177-PSMA-617 (Pluvicto) have only recently become FDA-approved, these agents, along with similar analogues, are already being evaluated in several clinical trials to determine the potential advantages of IA delivery (Table 3).

Figure 3 summarizes the paradigm of IA delivery of a radioembolic and a radiotheranostic, with the embolic drug becoming lodged within the vasculature and the radiopeptide entering the tumoral space to target cancer cells. The IA delivery method results in higher radiopeptide binding in tumors compared with IV delivery. This has been demonstrated in multiple clinical trials across a variety of cancer types and tissue locations.

Kratochwil et al. demonstrated improvement in therapeutic efficacy using the IA delivery of Y-90-DOTATOC to treat metastatic neuroendocrine tumors to the liver [40]. Y-90-DOTATOC is a somatostatin-receptor targeting ligand with a β-emitting radioisotope, making it functionally similar to Lu-177-DOTATATE (Lutathera). The IA delivery of DOTATOC was first proven with a Ga-68-labeled agent, and IA showed a 3.7-fold increase in tumor accumulation compared with IV dosing.

With the knowledge gained from the IA imaging study, Kratochwil et al. performed a therapeutic trial in 15 patients with unresectable neuroendocrine tumor metastasis to the liver [41]. The patients received hepatic artery-infused DOTATOC labeled with either Y-90 or Lu-177 after confirmation of efficacy with In-111-DOTATOC imaging. One patient had a complete response, eight had partial remission, and six had stable disease according to RECIST criteria. These results are promising, but without a comparison to IV therapy, it cannot be determined if the increased comparative uptake seen on imaging with IA delivery would lead to better tumor responses.

Thakral et al. performed a similar trial comparing IA to IV delivery of Lu-177-DOTATATE to neuroendocrine metastases [42]. A total of 29 patients were enrolled in the study; 15 patients received a single IA dose and the other 14 received a single IV dose. A threefold increase in tumor uptake was seen with IA compared to IV delivery, and no significant difference was seen in the absorbed dose to other healthy organs. Further, no patients in the IA group experienced any adverse events. These results mirror the findings from Kratochwil et al. [41].

Prostate cancer has been investigated as a target for both Y90 and IA delivery of radiotheranostics. Sayman et al. compared the ratio of the absorbed dose of Lu-177-PSMA in prostate lesions compared to healthy organs in the IA and IV delivery techniques [43]. Four patients were given IV treatment one week before receiving the same therapy through IA delivery. The patients were imaged with SPECT/CT to determine the total dose in lesions vs. healthy organs after each delivery technique. Improvements were seen in the ratio of radiation dose delivered to the dominant prostate lesion compared to the liver, bone marrow, healthy prostate, and whole body. When looking at the direct dominant lesion accumulation in IA vs. IV, only a 1.2-fold increase was observed. It was hypothesized that the initial IV treatment “stunned” the tumor and prevented a high uptake in the IA treatment. The critical advantage found in IA delivery was the lower dose present in healthy tissues compared to IV.

Two studies showed improvement in the accumulation of radiopharmaceutical drugs in meningioma with IA delivery. Verburg et al. compared IV and IA uptake of Ga-68-DOTATATE in four patients with inoperable meningiomas [44]. Compared to IV baseline uptake, patients averaged 2.7-fold more uptake in their respective tumors on PET/CT imaging after IA dosing. No change in toxicity was noted. More recently, Vonken et al. performed a similar study on four patients with meningiomas and compared IV and IA uptake of Lu-177-HA-DOTATATE [45]. Patients averaged a nearly five-fold increase in lesion accumulation of Lu-177 on SPECT/CT imaging with IA delivery. The technical success of the IA procedure was 100%, and no differences in adverse effects were found.

Averaging the reported ratio data from these five studies results in approximately three-fold more accumulation of IA delivered radiopharmaceuticals in tumor tissue compared to IV dosing. Summarized data and the calculated average are reported in Figure 4.

It is important to mention the recently finished LUTIA trial (official title: “Intra-arterial Lutetium-177-dotatate for Treatment of Patients with Neuro-endocrine Tumor Liver Metastases”). This trial employed a within-patient design where one liver lobe was treated IA and the opposite liver lobe received an IV treatment downstream from the IA infusion site [46]. The results of this trial in 26 patients have not been released yet, but they should give a great sense of the effectiveness of IA delivery of cancer receptor-targeted therapy.

## 6. Risks and Cost Comparison of IA vs. IV Delivery of Radiopharmaceuticals

Part of the attraction of IV-delivered radiopharmaceuticals is their low risk and low cost, as the procedure is essentially identical to any other IV-delivered drug. In comparison, IA procedures incur a much greater cost and have associated risks. Advancing a catheter into a deep arterial space requires a trained interventional radiologist, a large group of trained staff, intraprocedural imaging, and peri-procedural care. This adds additional costs and requires additional planning and utilization of critical procedural spaces. Additionally, image guidance procedures result in radiation exposure and additional risks compared to IV. Fluoroscopy-guided techniques require repeated X-ray exposures to visualize catheter advancement and position, but the total X-ray radiation dose is kept to a minimum as much as possible. IA procedures can rarely cause side effects, the most severe of which are vessel damage or hemorrhage.

Further, not all patients or tumors may be candidates for such a procedure. Some tumors may not have ideal vascular access or arterial components to make an IA procedure beneficial. Some tumor arteries can be tortuous, making it difficult to gain access and leading to poor target delivery. Embolics specifically need to be carefully placed in tumor-specific areas to prevent creating ischemia in healthy tissue. Additionally, leakage of embolics out of the tumor space could lead to accumulation and dosage in the lungs.

## 7. Alpha-Particle Therapy

Further driving the potential of radiotheranostics is the application of radioisotopes that decay by alpha (α)-particle emission in addition to radioisotopes that have only β-emission. Ra-223 dichloride is the only FDA-approved radiopharmaceutical that decays with α-particle emissions. The Ra-223 ionic chemical form binds most metastatic bone lesions with altered osteogenesis and has been used to treat many men with painful prostate bone metastases. All remaining FDA-approved radiopharmaceuticals rely on β-energy for radiation therapy. The β-decay occurs when a neutron converts to a proton and an electron and the high-energy electron (β-) is ejected from the nucleus. Radioisotopes decaying with β-emissions include I-131, Lu-177, and Y-90 and are used for a variety of reasons, including their half-life, purity, and ease of commercial production. One issue has arisen with β-emitting radiopharmaceuticals as our understanding has improved for applications in the preclinical and clinical settings: the penetration range of β-, while allowing for crossfire in non-targeted cancer cells, can deliver high and toxic radiation doses to the surrounding healthy tissues [47].

Alpha (α)-particles, which comprise two neutrons and two protons, are emitted from large unstable radioisotopes during decay. A comprehensive review of targeted α-particle therapy was recently published. It highlights not only current radiopharmaceuticals but also basic radiation biology [48]. Briefly, α-emitting radionuclides have not been widely used at the clinical level because of their low commercial availability and lack of pure α-emitting nuclides. Alpha (α)-particles are attractive from a cancer biology standpoint because of three major benefits compared to those that decay by only β-emissions: high linear energy transfer (LET), short penetration range, and efficiency in hypoxic environments.

Alpha (α)-particles have a linear energy transfer of 100 keV/µm compared to the 0.2 keV/µm of β-particles in tissue. The higher LET means a larger portion of the total radiation dose is delivered over an equal pathlength. Alpha (α)-particles can deliver up to 1000× more dose to cells than β-particles, even with the same number of radioactive decays. This high energy allows for double rather than single-strand DNA breaks, leading to increased cell death. Cancer cells can adapt to single-stranded DNA breaks and survive, but struggle when double-strand breaks occur. An incredible example of this occurred when β-resistance was overcome with α-particle therapy with Ac-225-PSMA [49]. As shown in Figure 5, Lu-177-PSMA was unable to debulk tumors or decrease PSA levels in this patient, but repeated doses of Ac-225-PSMA led to complete tumor eradication and a return to normal levels of PSA. Improvements in neuroendocrine cancer therapy are also seen when an α-particle emitter is used instead of β-radiation [50,51].

The short pathlength of α-particles is another advantage. Alpha (α)-particles deliver their energy over 40–90 µm of tissue, while β-penetrate 0.5–12 mm. Sensitive tissues near solid tumor locations, including prostate cancers, can be heavily irradiated during β-therapy. The range of α-particles is still large enough to cross multiple cell diameters, allowing for a local crossfire effect on non-targeted cells, but limiting the dose to healthy tissue [52].

Hypoxia, a hallmark of cancer, is notorious for causing resistance to a variety of cancer treatments [53]. Radioisotopes with primary β-emissions are no different and are not very effective to treat highly hypoxic tumors. The lower-energy radiation used in external-beam and β-therapies relies to a large degree on the formation of free radicals to induce cancer cell death. Conversely, α-particles rely solely on double-stranded breaks, thus making oxygen levels in the tumor environment irrelevant.

Many investigators have taken note of the advantages of using α-particles for cancer treatment [54]. Table 4 summarizes clinical trials utilizing α-particles for a multitude of cancer types. The variety of radionuclides and the total number of trials highlight the enthusiasm for using α-particles for therapy under the radiotheranostic paradigm.

## 8. Alternate Delivery Strategy for Alpha-Particles

With only one FDA-approved radiopharmaceutical with α-particle emissions, it is clear that α-emitting radiopharmaceuticals have not been successfully translated to clinical settings in spite of significant research in this area. There are a variety of radioisotopes with α-particle emissions that have unique physical and chemical properties that can be matched to their intended use, such as a specific half-life or desired chelator. Alternate delivery methods allow for the additional tailoring of the overall treatment strategy. The delivery methods of α-particles in the clinical setting include IA, intraperitoneal, and intravesical.

The strongest example supporting the use of a radioisotope with α-particle emissions delivered by an alternate route was from Kratochwil et al., who used Bi-213-DOTATOC-delivered IA to overcome previous resistance to Y-90-DOTATOC therapy [55]. Patients underwent an interventional procedure to deliver therapy to the hepatic artery, meaning the treatment went to most of the liver. Further, the leakage of therapy into the systemic system allowed for treatment of disseminated sites, which is shown in Figure 6. While this study did not compare IA to IV as closely as others, it represents a major advancement in understanding both α-particle therapy and IA delivery of radiopharmaceuticals.

The intraperitoneal delivery of α-particle radiopharmaceuticals can increase the targeting of peritoneal-confined disease while decreasing systemic toxicity. Meredith et al. showed the peritoneal retention of Pb-212-TCMC-Trastuzumab delivered directly into the peritoneal cavity [56]. In a dose-escalation phase 1 trial with 18 patients, single-dose therapy resulted in stable disease in several patients in the higher dose cohorts with no drug-related toxicities. This is a promising delivery strategy as it targets locally disseminated disease without total-body radiation.

Intravesical delivery for bladder cancer is a known concept that has shown synergy with radionuclide therapy. Autenrieth et al. showed the efficacy of Bi-213-labeled anti-EGFR antibody after intravesical delivery for the treatment of muscle-invasive bladder cancer [57]. The therapy was delivered through a urinary catheter into the bladder, allowing direct exposure to the tumor with no leakage into the bloodstream. The treatment was safe, with no toxicities reported, and the total dose delivered to the bladder wall was within tolerable limits. Further, the dose contribution of α, β-, and γ emissions to the bladder wall was calculated, showing that approximately 76% of the dose was from β- and 21% was from α. Even though α-particles have a much higher LET, only the decays that occur within microns of the wall will result in α-particle energy deposition, whereas β-particles have a range on the millimeter scale. The much larger range of β- results in multiple-fold more energy deposition events, outweighing the higher LET of α-particles.

Recently, external-beam radiation therapy combined with immunotherapy resulted in a systemic immune response against cancer, also known as the abscopal effect. Specifically, tumors irradiated with three fractionated doses (8 Gy each) showed synergy with anti-CTLA4 antibodies against both the irradiated tumor and a secondary untreated tumor [58]. Locoregionally delivered radiation therapy could be used to replicate the three-dose approach in cases where an external beam was contraindicated. Both embolic agents and biologically targeted radiopharmaceuticals can be delivered in this fashion, and some success has already been seen using low doses of repeated Lu-177 radiopeptide therapy to stimulate anti-cancer immune function when combined with immunotherapy [59].

Embolic radiopharmaceuticals mirror the external beam in that no cancer receptors are targeted, while IA-delivered radiopharmaceuticals targeting cancer receptors have the advantage of reaching higher tumor accumulation than via IV dosing with reduced dosing to normal tissues. Further research is needed with radiopharmaceuticals that decay quickly to recapitulate the fast and fractionated radiation therapies that are possible with an external beam. Table 5 outlines currently available short-lived α-particle emitters that could serve this purpose. Long-lived radioisotopes would give a gradually declining radiation treatment across their decay, thus abrogating the fractionated approach. Additionally, α-particle radiation’s inherent effects on cancer immunogenicity are becoming better understood, and it may be the most ideal radiation type for inducing synergy with the immune system [60].

Radioisotopes that decay by only α-particle emissions (no β-) are an attractive option but rare to find. One example is At-211 (7 h half-life), which is an α-emitter without any β-emission. However, it is important to note At-211’s source is a cyclotron equipped with an α-particle beam rather than a generator. Generators can be located at multiple sites or regional radiopharmacies and eluted by trained staff on an “as-needed” basis. This contrasts with a cyclotron, which must have a fixed location to produce the radioisotope followed by the transportation of the radioisotope to sites that do not have access to the cyclotron. This logistical issue makes it harder, but not impossible, for At-211 to be a longer-term option for fractionated therapy approaches. As an example, a Pb-212 generator can be eluted every 3 h to harvest Bi-212, with approximately 88% of the maximum activity being available at each elution. This allows not only for multiple patients to be treated per day, but for a single patient to reliably receive multiple treatments over a specific fractionated schedule. IA procedures can be easily performed on an in-patient or out-patient basis. The short half-life of the desirable radioisotopes means radiation dosing to the cancer in a short time interval, and patients become non-radioactive quickly as well, reducing radiation exposure to family members. Thus, it is an attractive strategy to use local generator-based radioisotope systems for the manufacturing of radiopharmaceuticals for immediate use with IA procedures. This method for a locoregional fractionated therapy strategy for cancer has the potential to achieve better outcomes, especially when combined with short-lived radioisotopes that emit α-particles during decay.

## 9. Conclusions

The future of radiotheranostics looks promising as more agents gain FDA approval and costs are covered by insurance reimbursement. The current process of obtaining reimbursement approval from the Centers for Medicare and Medicaid Services (CMS) is complicated and time-consuming, and often required before insurance companies will cover the costs of radiopharmaceuticals and imaging and the physicians’ fees. This is a significant problem that delays helping the maximum number of patients. Ideally, a streamlined process should allow for FDA approval to be simultaneous with reimbursement approvals. An additional problem is building the infrastructure, including installing cyclotrons to produce the radioisotopes, hot cells to safely handle the high levels of radiation, and radiopharmacies for cGMP manufacturing of the final radiopharmaceutical drugs. The process is often delayed by compliance approvals that involve multiple regulatory agencies as well as supply chain issues.

Off-target radiation damage to normal tissues is another significant problem with the current generation of therapeutic agents. For example, radiation doses to the kidneys from the normal excretion of Lutathera (Lu-177-DOTATATE) during the treatment of neuroendocrine tumors may limit future treatments beyond the initial four-dose schedule. The same may be true for patients with prostate cancer who are treated with Pluvicto (Lu-177-vipivotide tetraxetan) in terms of the dose to the salivary glands. Generally speaking, patients with high doses to normal tissues may not be eligible for additional rounds of therapy if they relapse and need additional therapy. The alternate approaches to delivering these radiopharmaceuticals by the IA route may help with this issue, especially if tumor uptake levels can be realized with lower total doses that translate into lower doses to normal tissues.

Cancer targeting with IV and/or IA delivery of radiopharmaceuticals addresses many of the drawbacks of external-beam radiation therapy. Continued research will yield advances in delivery strategies and radioisotope choices. Radioisotopes that decay by α-particle emissions cause double-stranded DNA breaks in tumors that are not easily repaired by cancer cells. When α-particle emitters are delivered to tumors in optimal dose schedules and combined with immunotherapy, a systemic immune response can be the result, which has a huge potential to improve patient outcomes.

## Figures and Tables

**Figure 1 pharmaceutics-15-01138-f001:**
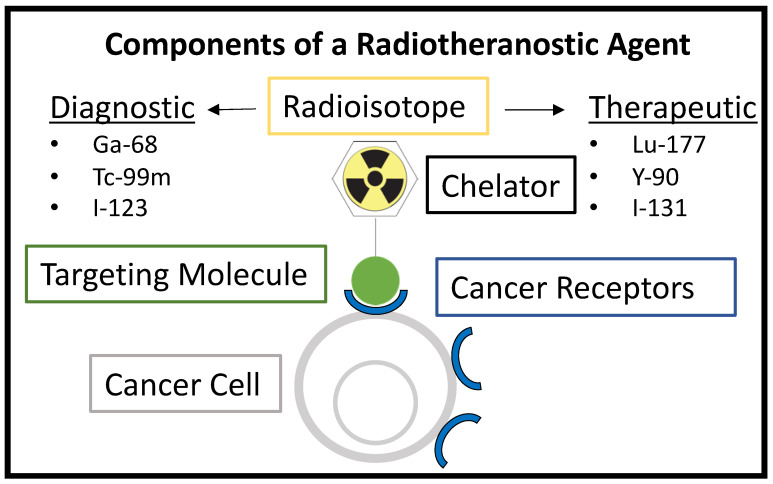
Components of a radiotheranostic, showing the same targeting molecule being used for either diagnostic or therapeutic applications depending on the radioisotope.

**Figure 2 pharmaceutics-15-01138-f002:**
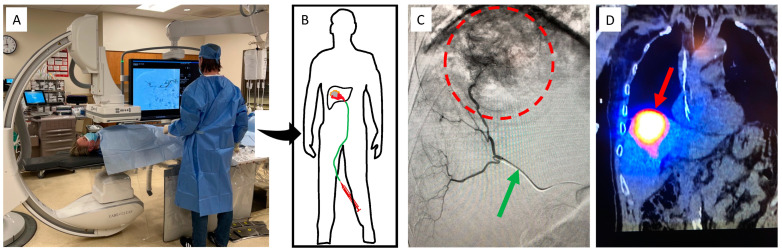
Overview of a fluoroscopy suite and Y90 procedure. Patients are placed on a bed underneath a fluoroscopy machine (**A**), which allows the physician to use image guidance to guide a catheter to the point of interest (**B**). Fluoroscopic imaging (**C**) allows visualization of a radio-opaque catheter (green arrow), which can be used to inject contrast dye that outlines the vessels feeding the tumor (red circle). Once in the proper location, therapeutic Y90 can then be delivered, and follow-up SPECT imaging (**D**) can confirm retention in the tumor.

**Figure 3 pharmaceutics-15-01138-f003:**
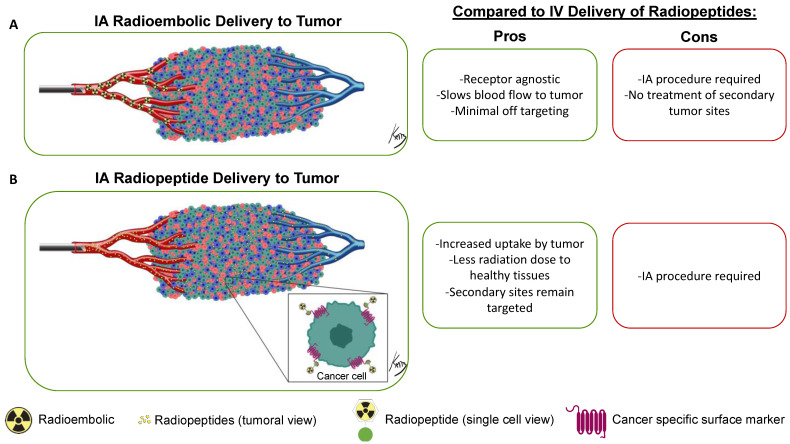
Overview of the IA delivery paradigm of radioembolics and radiopeptides. The radioembolic (**A**) becomes lodged in the smaller arterial vessels and will not cross over into the venous system. Conversely, radiopeptides (**B**) will enter the tumor space and bind to specific cancer receptors. Not all radiopeptides will be retained in the tumor and will instead travel systemically through the venous system. This allows for a higher accumulation of the radiopeptide in the tumor while still giving a systemic dose for satellite tumor-site treatment. In the tumor diagram, green cells are tumor cells expressing the specific marker, while red and blue cells are non-targeted tumor microenvironment cells. Figure 3 tumor vasculature: © Kevin Brennan 2023.

**Figure 4 pharmaceutics-15-01138-f004:**
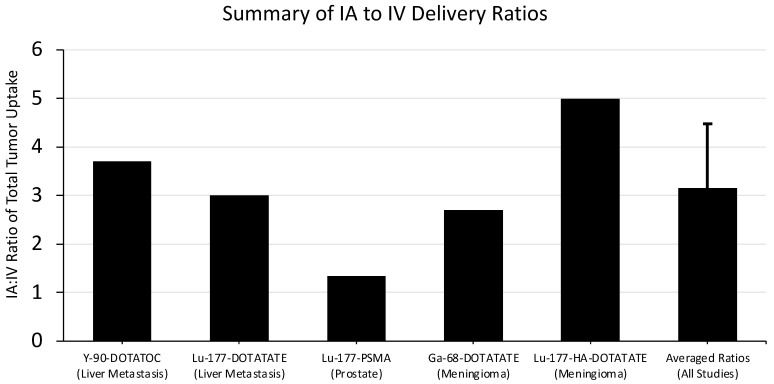
Summary of findings comparing IA and IV delivery of radiopharmaceuticals to tumors. The ratios from all studies were averaged, and the standard deviation of the average is reported here. The authors of the corresponding studies are as follows: Kratochwil et al. (Y-90-DOTATOC) [40], Thakral et al. (Lu-177-DOTATATE) [42], Sayman et al. (Lu-177-PSMA) [43], Verburg et al. (Ga-68-DOTATATE) [44], and Vonken et al. (Lu-177-HA-DOTATATE) [45].

**Figure 5 pharmaceutics-15-01138-f005:**
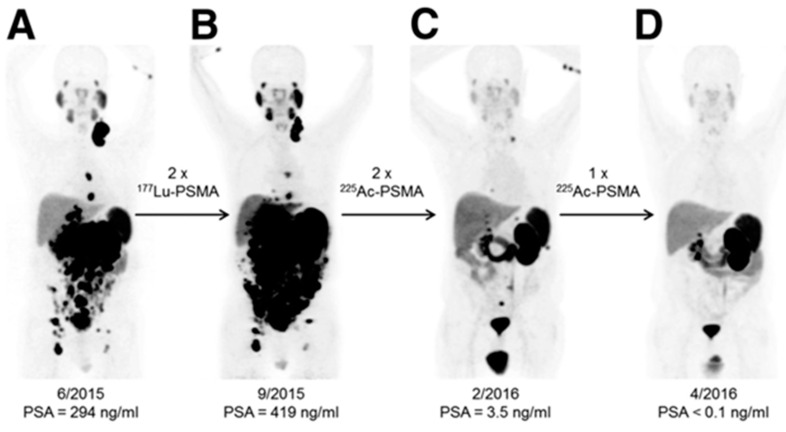
Alpha (α)-particle therapy overcomes β-resistant cancer. This imaging was performed using Ga-68-PSMA-11 PET/CT scanning. Kratochwil, C. et al. [49] reports, “In comparison to initial tumor spread (**A**), restaging after 2 cycles of β-emitting ^177^Lu-PSMA-617 presented progression (**B**). In contrast, restaging after second (**C**) and third (**D**) cycles of α-emitting ^225^Ac-PSMA-617 presented impressive response.” This research was originally published in *The Journal of Nuclear Medicine* [49].

**Figure 6 pharmaceutics-15-01138-f006:**
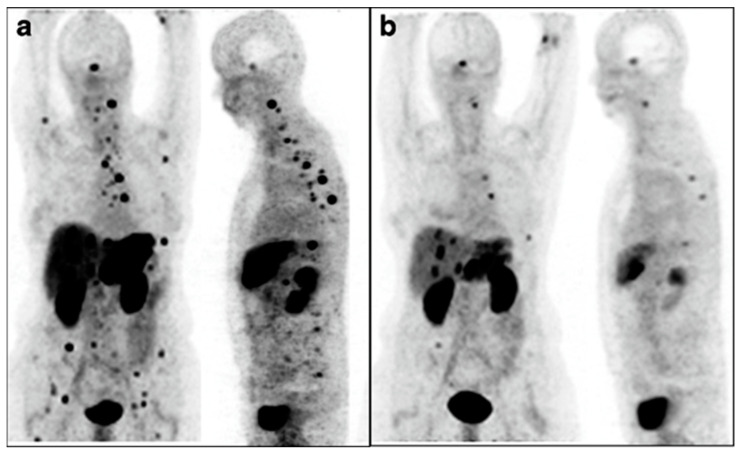
Intra-arterial Bi213-DOTATOC overcomes Y90-DOTATOC resistance. Ga-68-DOTATOC-PET imaging shows bulky liver disease and widespread lesions before treatment (**a**). Reduction in both primary liver and metastatic disease was seen on imaging six months after administration of Bi-213-DOTATOC into the common hepatic artery (**b**). This research was originally published in the *European Journal for Nuclear Medicine and Molecular Imaging* [55]. https://creativecommons.org/licenses/by/4.0/legalcode, accessed on 1 February 2023.

**Table 1 pharmaceutics-15-01138-t001:** FDA-approved Radiotheranostic Combinations.

Imaging Agent	Therapeutic Agent	Cancer Indication
I-123-Iodide (Sodium Iodide I-123)	I-131 Iodide (Hicon)	Hyperthyroidism and selected cases of thyroid carcinoma
I-123-iobenguane (MIBG, Adreview)	I-131 iobenguane (Azedra)	Pheochromocytoma and paraganglion
Ga-68-DOTATATE (Netspot) Ga-68-dotatoc Ga-68-gozetotide (Locametz and Illuccix) Cu64-DOTATATE (Detectnet)	Lu-177-DOTATATE (Lutathera)	Somatostatin-positive Neuroendocrine tumors
Ga-68-PSMA-11 F-18-piflufolastat (Pylarify)	Lu177- vipivotide tetraxetan (PSMA-617; Pluvicto)	PSMA-positive metastatic castration-resistant prostate cancer; for patients previously treated with androgen receptor pathway inhibition and taxane-based chemotherapy

**Table 2 pharmaceutics-15-01138-t002:** Other FDA-approved Radiation Therapeutics.

Imaging Agent	Therapeutic Agent	Cancer Indication
Tc-99m-medronate (MDP)	Sr-89 chloride (Metastron)	Bone metastases; areas of altered osteogenesis, typically increased osteoblastic activity
	Ra-223 dichloride (Xofigo)	Castration-resistant prostate cancer, symptomatic bone metastases, and no known visceral metastatic disease
Tc-99m-MAA	Y-90-loaded glass microspheres (Theraspheres) Y-90-resin microspheres (SIR-Spheres)	Hepatocellular carcinoma, off-label use for liver metastasis
In-111-Zevalin (the FDA removed the requirement for this scan prior to Y-90-Zevalin therapy in 2011)	Y-90 ibritiumomab tiuxetan (Zevalin)	Lymphoma
	P-32 Colloid	Cavitary metastases of cancer

**Table 3 pharmaceutics-15-01138-t003:** DOTATATE- and PSMA-Related Radiopeptides Delivered via IA in Clinical Trials.

Study Name [Reference Number]	Radiopharmaceutical Used	Phase	# of Participants
Intra-arterial Lutetium-177-dotatate for Treatment of Patients with Neuro-endocrine Tumor Liver Metastases (LUTIA) [NCT03590119]	Lu-177-DOTATATE	2/3	26
Lutathera in People with Gastroenteropancreatic (GEP), Bronchial, or Unknown Primary Neuroendocrine Tumors That Have Spread to the Liver [NCT04544098]	Lu-177-DOTATATE	1	10
Intra-arterial Hepatic (IAH) Infusion of Radiolabeled Somatostatin Analogs in GEP-NET Patients with Dominant Liver Metastases (LUTARTERIAL) [NCT04837885]	Lu-177-DOTATATE	2	20
Personalized PRRT of Neuroendocrine Tumors (P-PRRT) [NCT02754297]	Lu-177-Octreotate	2	300
Comparison of Hepatic Intra-arterial vs. Systemic Intravenous 68Ga-PSMA PET/CT for Detection of Hepatocellular Carcinoma [NCT05111314]	Ga-68-Gozetotide	1	10
Pharmacokinetics of IA and IV Ga68-PSMA-11 Infusion [NCT04976257]	Ga-68-PSMA-11	1	5

**Table 4 pharmaceutics-15-01138-t004:** All Active Trials Utilizing Available α-Particle Radionuclides for Cancer Therapy. Studies can be found by using the clinicaltrials.gov search tool with radionuclide terms formatted as in the following example: At-211 OR 211At OR Astatine-211.

α-Emitting Nuclide	Number of Trials	Cancers Targeted across All Nuclides
At-211	8	Myeloma Leukemia Thyroid Ovarian Neurological Non-malignant Neoplasm Colorectal Prostate Lung Bladder Gastric Breast Liver Bone
Ac-225	25	
Bi-213	2	
Th-227	4	
Pb-212	8	
Ra-223	104	

**Table 5 pharmaceutics-15-01138-t005:** Source and Decay Properties of Short-lived Radionuclides that Emit an α-Particle. https://periodictable.com/Isotopes/085.211/index3.full.dm.html, accessed on 4 February 2023.

Source (Parents)	Nuclide (Half-Life)	Decay Mode (%)	Daughters (Half-Life)	Daughters’ Decay Mode (%)	Other Daughters
Cyclotron: Bi-209(α,2n)At-211	At-211 (7.2 hr)	α (41.8%) 5.98 MeV	Bi-207 (32.9 yr)	β+ 1.37 MeV	
		EC (58.2%)	Po-211 (0.5 ms)	α (100%) 7.59 MeV	Pb-207 (Stable)
Generator: Ra-224 (3.6 d)→Fr-224→Ra-224-Rn-220→Po-216→Pb-212(10.6 h)→Bi-212 (Pb-212 or Bi-212 can be eluted)	Bi-212 (60.55 m)	β- (64.1%) 0.8 MeV	Po-212 (0.3 ms)	α (100%) 8.8 MeV	Pb-208 (Stable)
		α (35.9%) 6.1 MeV	Tl-108 (3.05 m)	β- (100%) 0.6 MeV	Pb-208 (Stable)
Generator: Ac-225 (10 d)→Fr-221→ At-217→Bi-213 (Bi-213 eluted)	Bi-213 (45.6 m)	β- (97.9%) 1.42 MeV	Po-213 (3.72 ms)	α (100%) 8.5 MeV	Pb-209 (3.2 h-100% β- 0.6 MeV)→Bi-209 (1.8 × 10^19^ yr)
		α (2.09%) 5.9 MeV	Tl-209 (2.2 m)	β- (100%) 3.97 MeV	Pb-209 (3.2 h-100% β- 0.6 MeV)→Bi-209 (1.8 × 10^19^ yr)
Generator: Rn-222 (3.8 d)→ Po-218 (3.1 m)→ Pb-214 (27 m; 100% β-, 1.02 MeV)→ Bi-214 (Pb-214 and Bi-214 are in equilibrium when recovered and used together, or Bi-214 may be purified and used separately from Pb-214)	Bi-214 (19.9 m)	β-(99.97%) 3.27 MeV	Po-214 (0.16 ms)	α 7.8 MeV	Pb-210 (22 yr)
		α (0.021%) 5.62 MeV	Tl-210 (1.3 m)	β- 5.4 MeV	Pb-210 (22 yr)
		β-, α (0.003%) 11.1 MeV	Pb-210 (22 yr)		

## Data Availability

Data is available from original references, clinicaltrials.gov, or other referenced web sites.

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
