# Peer review of "Intra-Arterial Delivery of Radiopharmaceuticals in Oncology: Current Trends and the Future of Alpha-Particle Therapeutics"

_pharmaceutics, 2023, doi:10.3390/pharmaceutics15041138_

Round 1
Reviewer 1 Report
The paper describes some of the more recent trends in the radiation therapy applied to cancer treatment. The review topic is interesting, and the English language and form are clear. However I have some suggestions and two concerns:
1) I am concerned about the small number of cited references, since only 42 papers are cited throughout the review. I think that, to write a comprehensive review, it is usually necessary to read and study a lot of Literature. Moreover, i think that a valuable review paper should provide the reader with a possibly high number of references that might stimulate further detailed study of the topic. I strongly suggest the authors to consider adding more citations to their work.
Among the possible citations, I suggest the following:
- Velikyan, I. 68Ga-Based Radiopharmaceuticals: Production and Application Relationship. Molecules 2015, 20, 12913–12943.
- Bolzati, C.; Dolmella, A. Nitrido technetium-99m core in radiopharmaceutical applications: Four decades of research. Inorganics 2020, 8, 3.
- Ailuno, G. et al. The Pharmaceutical Technology Approach on Imaging Innovations from Italian Research. Pharmaceutics 2021, 13, 8.
- Tepe, G. et al. Local intra-arterial drug delivery for prevention of restenosis - Comparison of the efficiency of delivery of different radiopharmaceuticals through a porous catheter. Investigative Radiology 2001, 36, 5, 245-249.
- Huang, R. et al. Strategies for Improved Intra-arterial Treatments Targeting Brain Tumors: a Systematic Review. Frontiers in Oncology 2020, 10.
- Sgouros, G. et al. Radiopharmaceutical therapy in cancer: clinical advances and challenges. Nature Reviews Drug Discovery 2020, 19, 589-608.
2) In the introduction, I suggest the Authors to mention the use of multi-step targeting strategies for the detection of different types of cancers. I suggest the Authors to cite the following papers:
- Altai, M. et al. Pretargeted Imaging and Therapy. J. Nucl. Med. 2017, 58, 1553–1559.
- Verhoeven, M. et al. Therapeutic Applications of Pretargeting. Pharmaceutics 2019, 11, 434.
- Paganelli, G.; Chinol, M. Radioimmunotherapy: Is avidin-biotin pretargeting the preferred choice among pretargeting methods? Eur. J. Nucl. Med. Mol. Imaging 2003, 30, 773–776.
- Pastorino, S. et al. Two Novel PET Radiopharmaceuticals for Endothelial Vascular Cell Adhesion Molecule-1 (VCAM-1) Targeting. Pharmaceutics 2021, 13.
2) I am also concerned about the poorness of the conclusions. Please improve this section. Conclusions should report final considerations and maybe also personal opinions of the Authors about the most promising future trends in the use of radiopharmaceuticals for cancer treatment, using intra-arterial administration.
3) On lines 37 and 165, the explanation of the abbreviation is repeated. Please remove the extended form on line 165.
4) Table 3 and 4: I think it would be very useful for the reader if the Authors add some reference to the trials relating the radiopharmaceutical used (Table 3) or the alpha-emitting nuclide (Table 4). I suggest the Authors to add the number identifying the clinical trials mentioned.
5) Figure 4: please insert the error bars in the graphs, thus the reader will be able to do rational comparisons between the findings of the different studies.
6) Table 4: I think that the last column is confusing, because it seems that all the mentioned alpha-emitting nuclides are under clinical trial for all the cancers mentioned in the last column. Please, subdivide the column indicating, for each nuclide, which types of cancer are targeted.
Reviewer 2 Report
The authors have presented various radiopharmaceutical delivery approaches in clinical scenarios. The authors have also highlighted efficacy of various literature reported IA and IV administered radiopharmaceuticals as well as discussed, the comparison of both the delivery approaches.
Comments:
The introduction is too brief. The authors should include an elaborate introduction to the review along with -highlighting the purpose of review.
Figure 4 - instead of using author names, it will be good to provide the radiopharmaceutical name for comparison. Authors can include more details in the figure legend.
Authors should include complete information on Alpha and beta particles in terms of LET X keV/µtissue range X µm, penetrations etc. with suitable references.
Authors should include more appropriate citations in section 7-Alpha particle therapy, section8- alternative strategies for Alpha particle therapy.
